# Microstructure and Magnetic Properties of Mn$_{55}$Bi$_{45}$ Powders Obtained by Different Ball Milling Processes

**Xiang Li [1], Dong Pan [1] 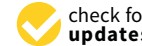, Zhen Xiang [2], Wei Lu [2,\*] and Dan Batalu [3]**

[1] School of Materials Science and Engineering, University of Shanghai for Science and Technology, Shanghai 200093, China; xiangli@usst.edu.cn (X.L.); pandong9109@163.com (D.P.)

[2] Shanghai Key Laboratory of Development & Application for Metal-Functional Materials, School of Materials Science and Engineering, Tongji University, Shanghai 200092, China; xiangzhen@tongji.edu.cn

[3] Materials Science and Engineering Faculty, University Politehnica of Bucharest, Bucharest 060042, Romania; dan.batalu@upb.ro

\* Correspondence: weilu@tongji.edu.cn; Tel.: +86-136-8184-5641

**Abstract:** Low-temperature phase (LTP) MnBi is considered as a promising rare-earth-free permanent magnetic material with high coercivity and unique positive temperature coefficient of coercivity. Mn$_{55}$Bi$_{45}$ ribbons with high purity of LTP MnBi phase were prepared by melt spinning. Then, Mn$_{55}$Bi$_{45}$ powders with different particle size were obtained by low-energy ball milling (LEBM) with and without added surfactant. The coercivity is enhanced in both cases. Microstructure characterization reveals that Mn$_{55}$Bi$_{45}$ powders obtained by surfactant assisted low-energy ball milling (SALEBM) have better particle size uniformity and show higher decomposition of LTP MnBi. Coercivity can achieve a value of 17.2 kOe and the saturation magnetization ($M_s$) is 16 emu/g when Mn$_{55}$Bi$_{45}$ powders milled about 10 h by SALEBM. Coercivity has achieved a maximum value of 18.2 kOe at room temperature, and 23.5 kOe at 380 K after 14 h of LEBM. Furthermore, Mn$_{55}$Bi$_{45}$ powders obtained by LEBM have better magnetic properties.

**Keywords:** Mn$_{55}$Bi$_{45}$ powders; low-energy ball milling; surfactant; particle size; coercivity; microstructure; magnetic properties

## 1. Introduction

Permanent magnets are indispensable components used in many vital areas including electric vehicles, medical equipment, high-energy product motors and generators, etc. [1,2]. Due to reserve crisis and high-cost of rare earth elements, there are a large amount of researches focusing on permanent magnets with reduced rare earth elements or rare earth-free permanent magnets [3–6].

Low-temperature phase (LTP) MnBi has been considered as a promising rare-earth-free magnet with unique magnetic properties [7–10]. However, it is difficult to obtain a single phase of LTP MnBi alloy, as Mn tends to segregate from the liquid phase below the peritectic temperature of 719 K [11,12]. Yang et al. [7] studied that Mn segregation can be reduced by melt-spinning process, in addition Mn$_{55}$Bi$_{45}$ can promote the formation of high purity LTP MnBi alloy. When MnBi powders mixed with other magnetic materials, the MnBi-based composites with high magnetic performance will be obtained [13,14]. It is critical to this method that the size of the MnBi powders is comparable to the size of their single magnetic domain (~500 nm) in order to maximize the loading of the soft magnetic phase [15,16]. Moreover, low-energy ball milling (LEBM) is the most successful method for fabricating MnBi powders with this size wheras too high mechanical energy may lead to the rapid decomposition of LTP MnBi [16]. The coercivity can be significantly enhanced by ball milling process, as it promotes the increase of lattice strain and dislocation density, as well as the reduction of particle size [17,18].

Xie et al. [16] reported that $Mn_{55}Bi_{45}$ powders were prepared by arc melting method and finished with LEBM at cryogenic temperature, coercivity reaches its maximum at 8 h of ball milling. Moreover, the record of magnetization 71.2 emu/g has been achieved via milling for 8 h, heat treated, and ball milled for extra half an hour process. Li et al. [19] studied the magnetic properties of anisotropic MnBi powders after LEBM performed in heptane, coercivity increases to 16.1 kOe at 5 h. The coercivity reaches a maximum of 16.2 kOe after 35 min of surfactant assisted low-energy ball milling (SALEBM) for MnBi particles obtained by Kanari et al. [20]. There is still a room to improve the coercvity of MnBi after ball milling. Furthermore, the contribution of SALEBM and the effect of surface defects on magnetization, coercivity are deserved to investigate further [20,21].

In our work, $Mn_{55}Bi_{45}$ powders with different particle size were obtained by low-energy ball milling, with and without surfactant. The size of $Mn_{55}Bi_{45}$ powders obtained by SALEBM is more uniform. Moreover, the room-temperature coercivity is enhanced in both cases, with an increase to 17.2 kOe by SALEBM and to 18.2 kOe by LEBM.

## 2. Materials and Methods

Commercial high purity manganese (99.99%) and bismuth (99.99%) (produced by Northeast Nonferrous Metals Market Co., Ltd., Shenyang, China) were used for obtaining a $Mn_{55}Bi_{45}$ ingot by induction melting in argon gas. The ingot was then cut into some parts and each part is about 5 g. Further, the ingot was used to obtain $Mn_{55}Bi_{45}$ ribbons by single-roller melt spinning with a tangential speed of 40 m/s. The ribbons were annealed for 30 min at 573 K in vacuum. The annealed ribbons were manually crushed into powders. Then powders were ground using a pestle and agate mortar, and sieved through a # 300 mesh resulting the particle size down to less than 48 μm. The sieved powders were milled using 2 methods: (1) milled for 2–14 h in heptane ($C_7H_{16}$) with the addition of 10 wt% oleic acid (OA, $C_{18}H_{34}O_2$), and (2) milled for 2–38 h only in heptane. The milling speed was 120 rpm. Zirconia balls of 3 mm diameter were used, with a ball to powder weight ratio of 10:1. The milled $Mn_{55}Bi_{45}$ powders were mixed with epoxy resin and aligned in a magnetic field of 1.8 T.

The crystallographic structure of the samples was examined by DX-2700 X-ray diffraction (XRD, Fangyuan Instrument Co., Ltd., Dandong, China) with Cu Kα radiation (λ = 1.5418 Å). The content of XRD phase was calculated by JADE 9. Quanta FEG 450 scanning electron microscopy (SEM, FEI, Hillsboro, OR, USA) was used to examine the size and morphology of the powders. The magnetic properties were measured by physical property measurement system (PPMS, Quantum Design Inc., San Diego, CA, USA) with an applied field up to 30 kOe.

## 3. Results and Discussion

### 3.1. High Purity of LTP MnBi Ribbons

In order to obtain high purity of LTP MnBi, $Mn_{55}Bi_{45}$ ribbons were prepared by melt spinning at a speed of 40 m/s and subsequently annealed in vacuum. The XRD patterns of $Mn_{55}Bi_{45}$ ribbon powders before annealing and annealed at 573 K for 30 min are shown in Figure 1. The weight percent of LTP MnBi phase is higher in annealed $Mn_{55}Bi_{45}$ ribbon powders. Before annealing, there are mainly Mn and Bi phases with little LTP MnBi in the ribbons. After annealing, the ribbons contain high purity of LTP MnBi over 90 wt% with a little Mn and Bi phases. Because during the formation of LTP MnBi, Mn tends to segregate from MnBi liquid at peritectic temperature [11].

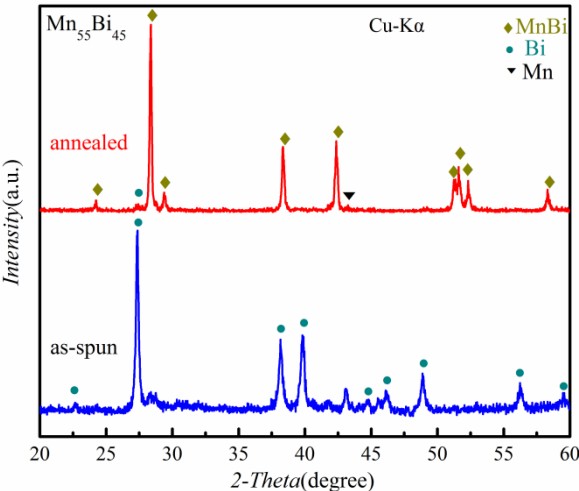

**Figure 1.** X-ray diffraction (XRD) patterns of $Mn_{55}Bi_{45}$ powders before and annealed at 573 K for 30 min.

The corresponding hysteresis loops of $Mn_{55}Bi_{45}$ powders before and after annealing are characterized and shown in Figure 2. The saturation magnetization ($M_s$) and coercivity are important magnetic parameters of a permanent magnet material which requires higher $M_s$ and coercivity. $M_s$ of the powders before annealing is 6.9 emu/g and achieves a higher value of 67 emu/g after annealing. The value of $M_s$ is in accordance with the increase of LTP MnBi after annealing presented in Figure 1. Since the annealed $Mn_{55}Bi_{45}$ ribbons still contain a little Mn and Bi phases shown in Figure 1. $M_s$ of annealed powders is close to the record of 71.2 emu/g achived by Xie et al. [16], but smaller than its theoretical value of 80 emu/g [22]. Coercivities of 7.4 kOe and 2.0 kOe are observed before and after annealing, respectively. Due to the sudden decrease in coercivity, it is desirable to obtain $Mn_{55}Bi_{45}$ powders with higher coercivity by different ball milling processes.

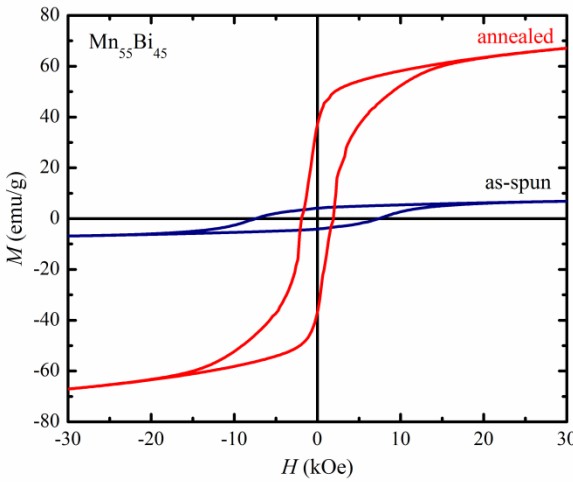

**Figure 2.** Hysteresis loops of $Mn_{55}Bi_{45}$ powders before and annealed at 573 K for 30 min.

*3.2. Microstructure of $Mn_{55}Bi_{45}$ Powders Obtained by Different Ball Milling Processes*

Figure 3 shows XRD patterns of $Mn_{55}Bi_{45}$ powders obtained by LEBM and SALEBM at different milling time. As shown in Figure 3a–e, the peak intensity of the LTP MnBi phase decreases, while characteristic peak of Bi and Mn phases gradually increase with prolonging the milling time, due to the partial decomposition of LTP MnBi during the ball milling processes [16]. The Bi phase has become the main phase of $Mn_{55}Bi_{45}$ powders in Figure 3e. The content of LTP MnBi phase decreases further when $Mn_{55}Bi_{45}$ powders is obtained by SALEBM, as shown in Figure 3c,e. As seen in Figure 3e, the reflection of Bi phase at 27.3° of $Mn_{55}Bi_{45}$ powders obtained by SALEBM at 14h significantly increases, and the

weight percentage of LTP MnBi decreases to 42 wt%. This indicates that the added surfactant (OA) leads to a higher decomposition of LTP MnBi.

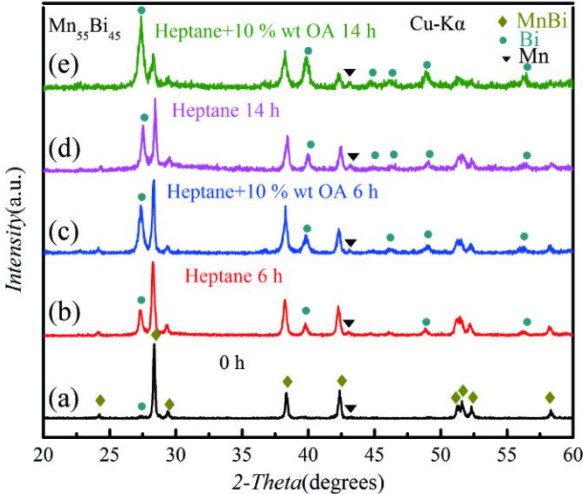

**Figure 3.** XRD patterns of $Mn_{55}Bi_{45}$ powders obtained by different ball milling processes at different milling time: (**a**) After annealing, (**b**) By LEBM at 6 h, (**c**) By SALEBM at 6 h, (**d**) By LEBM at 14 h and (**e**) By SALEBM at 14h.

In order to further explore the effect of different ball milling processes, microstructure of $Mn_{55}Bi_{45}$ powders was investigated. SEM images of the $Mn_{55}Bi_{45}$ obtained by SALEBM and by LEBM at different milling time are shown in Figures 4 and 5, respectively. The reduced size of $Mn_{55}Bi_{45}$ powders can be observed with the increase of time.

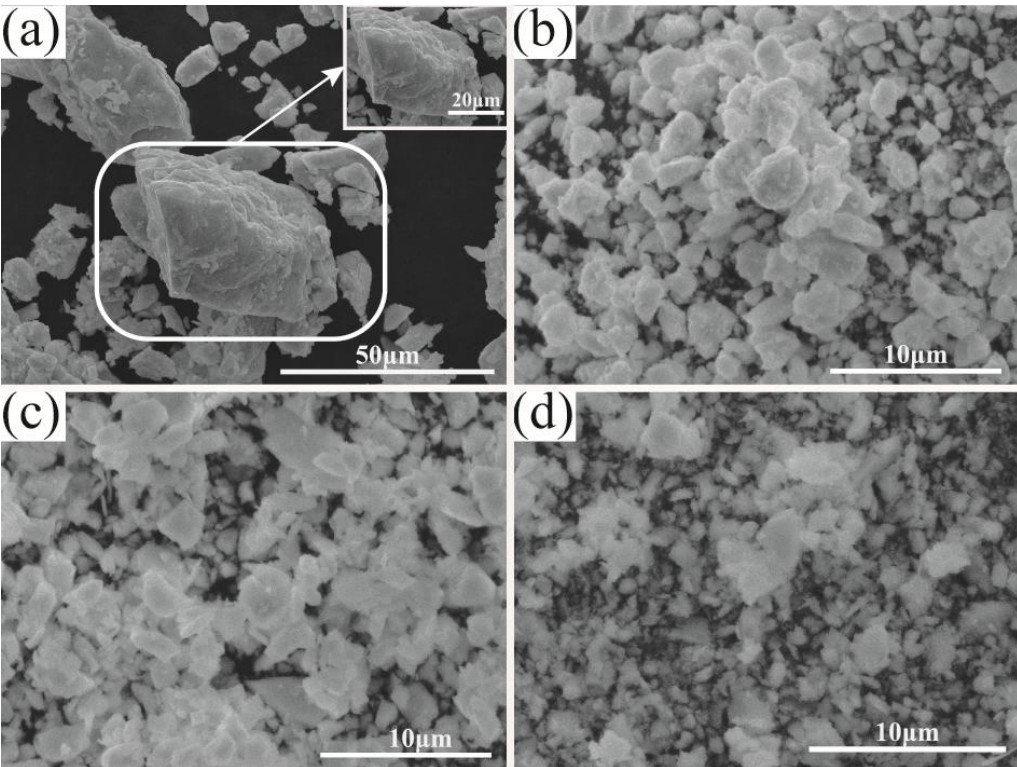

**Figure 4.** Scanning electron microscopy (SEM) images of (**a**) Crushed $Mn_{55}Bi_{45}$ powders and $Mn_{55}Bi_{45}$ powders obtained by SALEBM at (**b**) 4 h, (**c**) 10 h, (**d**) 14 h.

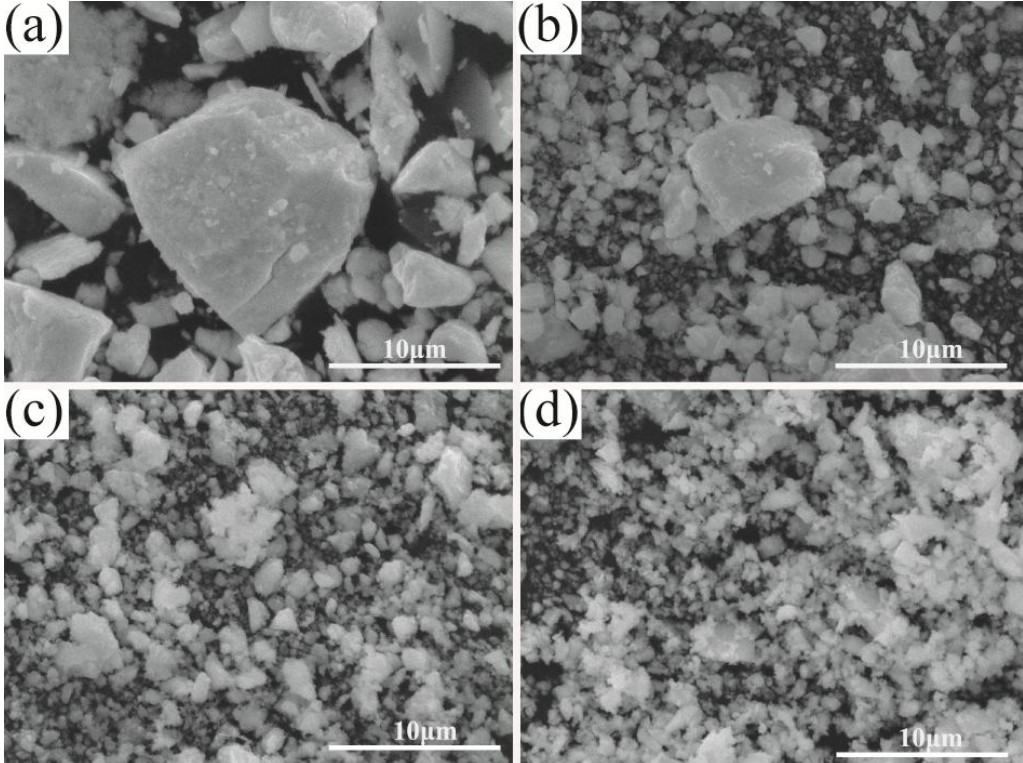

**Figure 5.** SEM images of Mn$_{55}$Bi$_{45}$ powders obtained by LEBM at (**a**) 4 h, (**b**) 10 h, (**c**) 14 h, (**d**) 34 h.

As seen in Figures 4a and 5a, coarse particles with small flaky particles on the their surfaces are mainly observed. As shown in Figure 4a, Mn$_{55}$Bi$_{45}$ ribbons was manually crushed and the size of resulted powders range between 1–40 μm before milling. The size distribution of SALEBM at 4 h is ~2 μm in Figure 4b. Figure 4b–d shows that the size of Mn$_{55}$Bi$_{45}$ powders obtained by SALEBM was reduced to ~4 μm, ~2 μm, and ~0.8 μm after 4, 10 and 14 h, respectively. The particles after 10 h of SALEBM have a size distribution of ~1 μm, as shown in Figure 4c. The size of powders is uniform and the shape is very regular. The size of Mn$_{55}$Bi$_{45}$ powders after 10 h of SALEBM is more homogenous. Agglomeration of the powders forms due to the surfactant molecules exist in the powders, as shown in Figure 4b,c. As shown in Figure 4d, Mn$_{55}$Bi$_{45}$ powders have less agglomerates, because the surfactant molecules have been adsorbed at the surface of powders [20]. As seen in Figure 5, the size of Mn$_{55}$Bi$_{45}$ powders obtained by LEBM (without surfactant) was reduced to ~14 μm, ~8 μm, and ~2.5 μm after 4, 10 and 14 h, respectively. The size distribution of LEBM is ~3 μm and ~2 μm after 10 and 14 h in Figure 5b,c, respectively. The size of powders is different, and the shape of particles is polyhedral which is very irregular. As shown in Figure 5d, the size of Mn$_{55}$Bi$_{45}$ powders obtained at 34 h by LEBM is ~0.7 μm which tends to equalize. This size becomes smaller and more uniform which cannot further be reduced, the coarse powders finally reduced down to finer particles. Compared Figure 4 with Figure 5, at the same milling time (<14 h), the powders size of SALEBM is smaller than the size of LEBM. Therefore, the added surfactant (OA) helps Mn$_{55}$Bi$_{45}$ powders to be milled with a higher size refinement and uniformity at shorter time, with the drawback of a higher decomposition of LTP MnBi shown in Figure 3.

### 3.3. Magnetic Properties of Mn$_{55}$Bi$_{45}$ Powders Obtained by Different Ball Milling Processes

To further investigated the contribution of different ball milling processes, the magnetic properties of Mn$_{55}$Bi$_{45}$ powders were measured. Hysteresis loops and variation of coercivity, $M_s$ with milling time of Mn$_{55}$Bi$_{45}$ powders obtained by SALEBM and LEBM are shown in Figures 6 and 7, respectively. $M_s$ gradually decreases with the increase of time, this is in agreement with the decreasing weight

percentage of the LTP MnBi phase in Figure 3. The coercivity is simultaneously enhanced from 2.0 kOe at 0 h to a maximum due to the smaller particle size is less than the single domain size [23]. It is also important for a permanent magnet to achieve higher ratio of remanence ($M_r$) to $M_s$ ($M_r/M_s$) and magnetic energy product $(BH)_{max}$. The change of the main magnetic properties such as $M_s$, coercivity, $M_r/M_s$, and $(BH)_{max}$ of $Mn_{55}Bi_{45}$ powders obtained by SALEBM and LEBM after different milling time are listed in Table 1.

As shown in Figure 6 and Table 1, $Mn_{55}Bi_{45}$ powders show a maximum coercivity of 17.2 kOe when milled 10 h by SALEBM. Coercivity achieves a maximum value of 18.2 kOe when $Mn_{55}Bi_{45}$ powders milled for 14 h by LEBM, as seen in Figure 7 and Table 1. The room-temperature coercivity is enhanced in both cases. The particle refinement and the increase of the stresses or the defects during ball milling process increases the coercivity [17,18]. This conclusion is in accordance with the refined microstructure of $Mn_{55}Bi_{45}$ powders shown in Figures 4 and 5. Figure 7b and Table 1 show that coercivity fast increased to 13.1 kOe after 2 h of milling by LEBM, achieving a higher value than 5.6 kOe by SALEBM at 2 h. Since coercivity depends on the microstructure, different sizes and irregular shapes of $Mn_{55}Bi_{45}$ powders shown in Figure 5 are beneficial to coercivity. Hence, LEBM is a more favorable method to significantly improve the coercivity. As shown in Figure 6 and Table 1, $M_s$ drops to 43, 19, 16, and 9 emu/g when milled by SALEBM for 2, 6, 10, and 14 h, respectively. On the contrary, as shown in Figure 7, $M_s$ drops to 52, 32, 20, 16, and 10 emu/g when milled by LEBM for 2, 6, 10, 14, and 18 h, respectively. The decomposition of the LTP MnBi will reduce the magnetization of the MnBi alloy [24]. $Mn_{55}Bi_{45}$ powders after ball milling with long time contain a large volume fraction of nonmagnetic Bi phase, hence a lower $M_s$ is expected and confirmed by measurements, which reduces to 1 emu/g after 38 h of SALEBM shown in Figure 7. Figures 6 and 7 indicate that $M_s$ of $Mn_{55}Bi_{45}$ powders decreases more rapidly when milled by SALEBM, this result is in agreement with the content of LTP MnBi phase decreases more rapidly (Figure 3).

As shown in Table 1, $M_r/M_s$ of $Mn_{55}Bi_{45}$ powders increases from 55.1% at 0 h to 91.1% at 6 h of SALEBM and 90.0% at 4h of LEBM. It reveals that most of milled powders are aligned to the c-axis. However, $M_r/M_s$ of SALEBM decreases more quickly indicating the powders of LEBM are better aligned to the c-axis [9]. $(BH)_{max}$ increase from 2.54 MGOe before milling to 4.2 MGOe after 2 h of SALEBM. $(BH)_{max}$ is 7.1 MGOe after 2 h of LEBM, which is equal to the value of 7.1 MGOe after 7 h of grinding reported by Yang et al. [22]. Moreover, all $(BH)_{max}$ of $Mn_{55}Bi_{45}$ powders obtained by LEBM are bigger. Therefore, $Mn_{55}Bi_{45}$ powders obtained by LEBM have better magnetic properties.

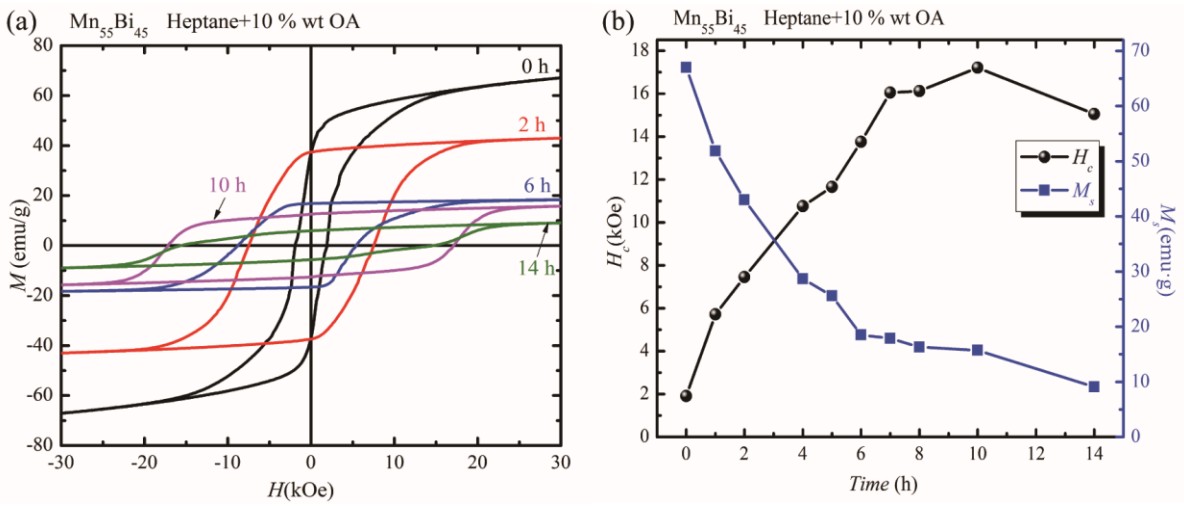

**Figure 6.** (**a**) Hysteresis loops and (**b**) Variation of coercivity, $M_s$ with milling time of $Mn_{55}Bi_{45}$ powders obtained by SALEBM.

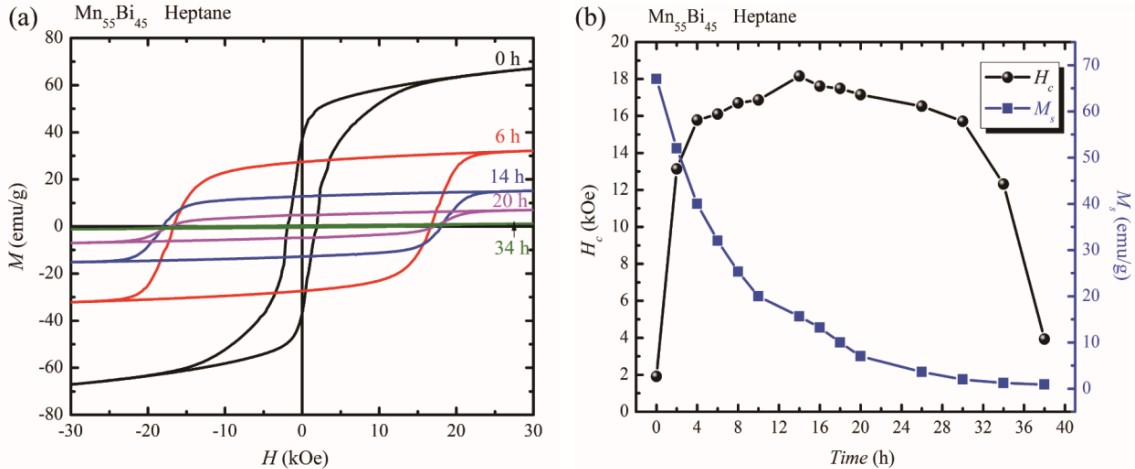

**Figure 7.** (**a**) Hysteresis loops and (**b**) Variation of coercivity, $M_s$ with milling time of Mn$_{55}$Bi$_{45}$ powders obtained by LEBM.

**Table 1.** Change of the main magnetic properties of Mn$_{55}$Bi$_{45}$ powders obtained by different ball milling processes after different milling time.

| Sample | $M_s$ (emu/g) | $H_c$ (kOe) | $M_r/M_s$ (%) | $(BH)_{max}$ (MGOe) |
|---|---|---|---|---|
| Heptane + 10 wt% OA (2h) | 43 | 7.5 | 86.9 | 4.2 |
| Heptane (2h) | 52 | 13.1 | 89.9 | 7.1 |
| Heptane + 10 wt% OA (4h) | 27 | 10.8 | 88.3 | 2.0 |
| Heptane (4h) | 40 | 15.8 | 90.0 | 4.1 |
| Heptane + 10 wt% OA (6h) | 19 | 13.8 | 91.1 | 0.9 |
| Heptane (6h) | 32 | 16.1 | 85.3 | 2.3 |
| Heptane + 10 wt% OA (10h) | 16 | 17.2 | 79.4 | 0.5 |
| Heptane (10h) | 20 | 16.9 | 84.6 | 1.0 |
| Heptane + 10 wt% OA (14h) | 9 | 15.1 | 65.1 | 0.1 |
| Heptane (14h) | 16 | 18.2 | 84.5 | 0.3 |

Mn$_{55}$Bi$_{45}$ powders obtained by LEBM for 14 h are selected to measure the high temperatures magnetic properties of MnBi powders. Figure 8 shows hysteresis loops at 300 K, 325 K, 350 K, 380 K and variation of coercivity, $M_s$ with temperature. $M_s$ decreases with with the incease of temperature. Coercivity of Mn$_{55}$Bi$_{45}$ powders increase to 23.5 kOe at 380 K. With the increase of temperature, coercivity rapidly increases exhibiting an unusual positive temperature coefficient of coercivity. This positive temperature coefficient of coercivity is related to the magnetocrystalline anisotropy of MnBi [7,25], owing to the non-uniform change of lattice constants with temperature [7].

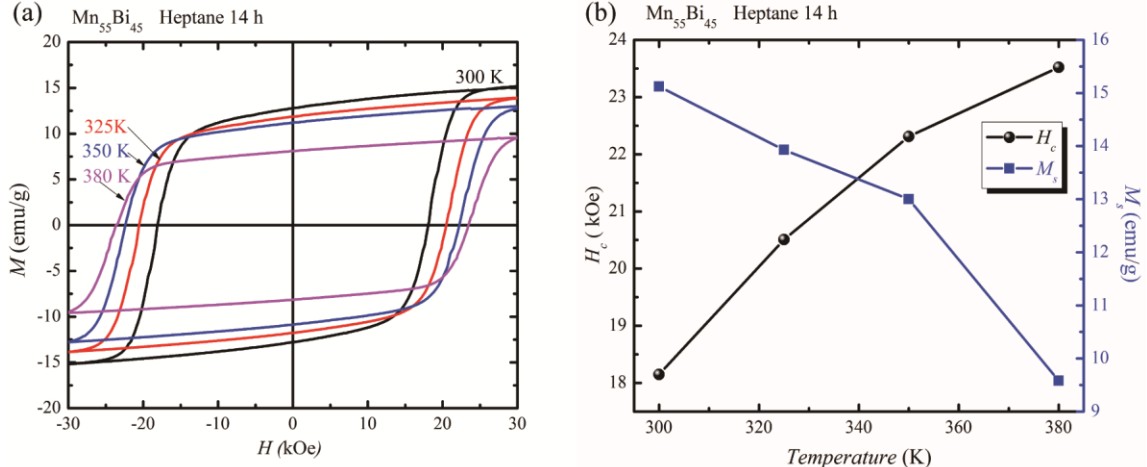

**Figure 8.** (**a**) Hysteresis loops at 300 K, 325 K, 350 K, 380 K and (**b**) Variation of coercivity, $M_s$ with temperature of $Mn_{55}Bi_{45}$ powders obtained by LEBM for 14 h.

## 4. Conclusions

In conclusion, $Mn_{55}Bi_{45}$ ribbons with high purity of LTP MnBi were successfully prepared. Furthermore, $Mn_{55}Bi_{45}$ powders with different particle sizes were obtained by low-energy ball milling with and without surfactant (OA). $M_s$ of powders obtained by both processes gradually decreases with the increase of time. $Mn_{55}Bi_{45}$ powders obtained by SALEBM have a higher size refinement and uniformity at shorter time, but with the drawbacks of higher decomposition of LTP MnBi and lower $M_s$. LEBM is a more favorable method to significantly improve the coercivity. Coercivity achieves a maximum value of 18.2 kOe at room temperature and 23.5 kOe at 380 K when MnBi powders milled 14 h by LEBM, without surfactant addition. Moreover, $Mn_{55}Bi_{45}$ powders obtained by LEBM have better magnetic properties. This approach can be applied to obtain MnBi-based composites with high magnetic performance.

**Author Contributions:** Conceptualization, W.L.; methodology, X.L. and X.Z.; validation, W.L.; formal analysis, D.P.; investigation, X.L. and W.L.; resources, X.L. and W.L.; data curation, X.L. and D.P.; writing—original draft preparation, X.L.; writing—review and editing, D.P. and D.B.; visualization, D.P. and X.Z.; supervision, X.Z. and D.B.; project administration, W.L.; funding acquisition, X.L., W.L. and D.B.

**Funding:** This work was supported by the Natural Science Foundation of Shanghai (No. 17ZR1419700), the National Natural Science Foundation of China (Grant No. 51671146), the Fundamental Research Funds for the Central Universities (No. 2016117), Shanghai Key Laboratory of Impression Evidence (No. 20163003), and by a grant of the Romanian Ministry of Research and Innovation, CCCDI-UEFISCDI, project number PN-III-P3-3.1-PM-RO-CN-2018-0113/17/2018, within PNCDI III.

**Conflicts of Interest:** The authors declare no conflict of interest.

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
