# Peer review of "Microstructure and Magnetic Properties of Mn55Bi45 Powders Obtained by Different Ball Milling Processes"

_metals, doi:10.3390/met9040441_

Round 1

Reviewer 1 Report

General comments:

1) Extensive editing of the English language (grammar and spelling) should be conducted for this paper.

2) The paper is missing a discussion section

3) This paper’s work and results are very similar to the work done by Xie et al. (reference [16]) except that it is conducted in at room temperature in heptane with and without surfactant (oleic acid) instead of at cryogenic temperatures. A thorough comparison of the results from this paper and previous work with emphasis on the unique contribution of this work to the general literature would greatly strengthen this paper.

Introduction

1) The authors claim, based on previously published research, that it is critical to obtain a particle size comparable to the size of the MnBi magnetic domain (~500 nm) (lines 39-40). Please expand on why it is so critical to reach this particle size. The authors were able to reduce the particle size of their powder down to ~700 - 800 nm. Is this close enough to the critical particle size to reach the desired properties?

2) In general, ball milling increases (not reduces) dislocation density (see line 43).

Materials and Methods

1) Typically the manufacturer and the location of the manufacturer is provided when discussing materials used.

2) Please explain by what method the ribbons were “ground down”.

3) Typically the manufacturer and model number of each piece of equipment is provided in the methodology section.

Results

1) Lines 72-73: “The intensity of LTP MnBi phase of annealed Mn55Bi45 powders were higher than the powders before annealing” is contradicted by Figure 1. Please clarify.

2) Line 75: How was the weight percent of each phase measured? XRD – Rietveld analysis? Please describe this analysis in detail in the “Materials and Methods” section.

3) Please increase the symbol size for Figures 1 and 3. It is difficult to make out in a print version of this article.

4) Figure 3a is not mentioned in the text.

5) Quantitative analysis of the particles size distribution would greatly strengthen the quality of the article.

6) The two sentences contained in lines 111 – 113 contradict one another.

7) The shapes of the two sets of powders, milled with or without surfactant, look similar to me (Figures 4 and 5). Quantitative analysis or measurement of the shapes of the particles (via aspect ratio, sphericity, etc.) would greatly strengthen the argument.

8) Lines 127-128: XRD does not provide information on saturation magnetization. Please clarify this sentence.

9) Lines 128-129: “The coercivity is simultaneously enhanced from 2.0 kO3 at 0 h to a maximum, then decreases due to the particle size may be smaller than the single domain size [23].” You reported earlier in the text that the single domain size was ~500 nm but that your particle size only reached a minimum of 600 - 700 nm. Please clarify this statement.

10) Please provide for each of the magnetic properties discussed in section 3.3: a brief introduction, equation for calculation (as appropriate) and description as to why each is important (and potentially what values are acceptable) for the design of a permanent magnetic material.

11) Figures 6a and 7a are not really discussed in the text. Are they needed to support the findings of the article?

12) Line 136: When it comes to ball milling, there is a lot of confusion because the phrases ‘grain size’ and ‘particle size’ are often used interchangeably (but are not the same thing!). Use caution when using both in the same paper.

13) Figures 6 and 7 reproduce poorly when printed in black and white. Consider adding a reference to ‘the web version for color’ to the caption of these figures.

14) Figures 6b and 7b: it is recommended that arrows be placed near each line to further clarify which line goes to which y-axis.

15) Lines 164 - 165: The English here makes it sound like you ball milled at elevated temperatures instead of measuring the magnetic properties at elevated temperatures. Please clarify.

Conclusions

1) The improvement to coercivity from ball milling is discussed but not the loss in saturation magnetization.

Author Response

The authors appreciate the suggestions and comments made by the reviewer. Modifications were made and highlighted in the revised version, according to the reviewer’s comments. In the following, the alterations made in the revision are summarized:

1. General comments:

1) Extensive editing of the English language (grammar and spelling) should be conducted for this paper.

2) The paper is missing a discussion section.

3) This paper’s work and results are very similar to the work done by Xie et al. (reference [16]) except that it is conducted in at room temperature in heptane with and without surfactant (oleic acid) instead of at cryogenic temperatures. A thorough comparison of the results from this paper and previous work with emphasis on the unique contribution of this work to the general literature would greatly strengthen this paper.

Reply: 1) English language (grammar and spelling) has been improved

2) The discussion section has been supplied in “Results and Discussion”.

3) Xie et al. put their effort to improve the magnetization. In our paper, we use melt spinning to achive a high magnetization, finally we want a higher coercivity after ball milling and investigate the contribution of different ball processes on magnetization, coercivity. A comparison of the results from this paper and the results by Xie et al. (reference [16]) has been added.

 Lines 44-47 Xie et al.[16] reported that Mn55Bi45 powders were prepared by arc melting method and finished with LEBM at cryogenic temperature, coercivity reaches its maximum at 8 hours of ball milling. Moreover, the record of magnetization 71.2 emu/g has been achieved via milling for 8 hours, heat treated, and ball milled for extra half an hour process."

Lines 89-90 ”Ms of annealed powders is close to the record of 71.2 emu/g achived by Xie et al. [16]"

2. Introduction:

1) The authors claim, based on previously published research, that it is critical to obtain a particle size comparable to the size of the MnBi magnetic domain (~500 nm) (lines 39-40). Please expand on why it is so critical to reach this particle size. The authors were able to reduce the particle size of their powder down to ~700 - 800 nm. Is this close enough to the critical particle size to reach the desired properties?

2) In general, ball milling increases (not reduces) dislocation density (see line 43).

Reply: 1) The reason ”in order to maximize the loading of the soft magnetic phase(lines 40) has been given. Secondly, yes, ~700 - 800 nm is close enough to the critical particle size to reach the desired properties due to the smaller particle size is less than the single domain size.(lines 140-141). (smallest)

2) Line 43, “reduction were changed to “increase

3. Materials and Methods

1) Typically the manufacturer and the location of the manufacturer is provided when discussing materials used.

2) Please explain by what method the ribbons were “ground down”.

3) Typically the manufacturer and model number of each piece of equipment is provided in the methodology section.

Reply: 1) The manufacturer and the location of the manufacturer is provided ”produced by Shenyang Northeast Nonferrous Metals Market Co., Ltd.(lines 60)

2) The ribbons were ground using a pestle and agate mortar (lines 65)

3) X-ray diffraction (XRD, DX-2700X). Physical property measurement system manufactured by Quantum Design, Inc. (lines 75)

4. Results

1) Lines 72-73: “The intensity of LTP MnBi phase of annealed Mn55Bi45 powders were higher than the powders before annealing” is contradicted by Figure 1. Please clarify.

Reply: “The intensity of LTP MnBi phase of annealed Mn55Bi45 powders were higher than the powders before annealing” was changed to The weight percent of LTP MnBi phase is higher in annealed Mn55Bi45 powders.

2) Line 75: How was the weight percent of each phase measured? XRD – Rietveld analysis? Please describe this analysis in detail in the “Materials and Methods” section.

Reply: The content of XRD phase was calculated by JADE 9.  was added in the “Materials and Methods” section.

3) Please increase the symbol size for Figures 1 and 3. It is difficult to make out in a print version of this article.

Reply:The symbol size for Figures 1 and 3 was increased to 28

4) 
Figure 3a is not mentioned in the text.

Reply: In Line 99, in Figure 3b-e was changed to in Figure 3a-e.

5) Quantitative analysis of the particles size distribution would greatly strengthen the quality of the article.

Reply: The size distribution of SALEBM at 4 h is ~2 μm in Figure 4b. is added. (Lines 120-121)

The particles after 10 h of SALEBM have a size distribution of ~1 μm, as shown in Figure 4c. is added. (Lines 122-123)

The size distribution of LEBM is ~3 μm and ~2 μm after 10 and 14 h in Figure 5b,c , respectively.  is added. (Lines 129-130)

6) The two sentences contained in lines 111 – 113 contradict one another.

Reply: The front sentence has left out "as shown in Figure 4b,c"

7) The shapes of the two sets of powders, milled with or without surfactant, look similar to me (Figures 4 and 5). Quantitative analysis or measurement of the shapes of the particles (via aspect ratio, sphericity, etc.) would greatly strengthen the argument.

Reply: As seen in Figure 4a and 5a, coarse particles with small flaky particles on the their surfaces are mainly observed. is added. (Lines 118-119)

8) Lines 127-128: XRD does not provide information on saturation magnetization. Please clarify this sentence.

Reply: It was changed to this is in agreement with the decreasing weight percentage of the LTP MnBi phase in Figure 3.

9) Lines 128-129: “The coercivity is simultaneously enhanced from 2.0 kOe at 0 h to a maximum, then decreases due to the particle size may be smaller than the single domain size [23].” You reported earlier in the text that the single domain size was ~500 nm but that your particle size only reached a minimum of 600 - 700 nm. Please clarify this statement.

Reply: Lines 140-141: ”...a maximum, then decreases due to the particle size may be smaller than the single domain size [23]. was changed to a maximum due to the smaller particle size is less than the single domain size.

10) Please provide for each of the magnetic properties discussed in section 3.3: a brief introduction, equation for calculation (as appropriate) and description as to why each is important (and potentially what values are acceptable) for the design of a permanent magnetic material.

Reply: Lines 88-89: The saturation magnetization (Ms) is an extremely important magnetic parameter of a permanent magnet material which requires a higher Ms.

Lines 141-142: It is also important for a permanent magnet to achieve higher ratio of remanence (Mr) to Ms (Mr/Ms) and magnetic energy product (BH)max.

11) Figures 6a and 7a are not really discussed in the text. Are they needed to support the findings of the article?

Reply: We think that it is really important to present Figures 6a and 7a because these Figures are the original measurement data in order to support veracity of this article.

12) Line 136: When it comes to ball milling, there is a lot of confusion because the phrases ‘grain size’ and ‘particle size’ are often used interchangeably (but are not the same thing!). Use caution when using both in the same paper.

Reply: All phrases grain size were changed to particle size.

13) Figures 6 and 7 reproduce poorly when printed in black and white. Consider adding a reference to ‘the web version for color’ to the caption of these figures.

Reply: Arrows and Labels was added in Figures 6 and 7 .

14) Figures 6b and 7b: it is recommended that arrows be placed near each line to further clarify which line goes to which y-axis.

Reply: Legend was added in Figures 6b, 7b and 8b

15) Lines 164 - 165: The English here makes it sound like you ball milled at elevated temperatures instead of measuring the magnetic properties at elevated temperatures. Please clarify.

Reply: We have corrected this sentence to measure the high temperatures magnetic properties of MnBi powders

5. Conclusions

1) The improvement to coercivity from ball milling is discussed but not the loss in saturation magnetization.

Reply: Ms of powders obtained by both processes gradually decreases with the increase of time.  and lower Ms 

Reviewer 2 Report

In the paper entitled "Microstructure and Magnetic Properties of Mn55Bi45 Powders Obtained by Different Ball Milling Processes", the authors give a comprehensive overview of the structural and magnetic properties a MnBi alloy after different treatments. These treatments include 2 different Ball milling processes (both applied for different times) and an investigation of the magnetic properties as function of temperature.

I found the manuscript of good quality with a good title and abstract that cover te contents well. All conclusions are supported by the data, which itself is presented in clear figures and tables.

Before I can recommend to accept this paper for publication I have to minor remarks that should be addressed.

Firstly, on line 170 of page 7 the authors state that "the positive temperature coefficient of coercivity is supposed to be related to the magnetocrystalline anisotropy of MnBi"
I found this statement quite vague as "supposed to be related" does not say anything. Can the authors explain how it is related? Is it for instance because the anisotropy constant itself increases as function of temperature or are there other factors at play? Please clarify how this can be explained.

Secondly, although the manuscript is well written, some sentences have strange grammatical structures. Can the authors let the manuscript get proofreader by a native speaker (or does this journal do this upon acceptance) to edit the text to proper English?

In summary, I found this an interesting paper that can be useful to the community and I recommend a minor revision.

p { margin-bottom: 0.1in; line-height: 115%; }

Author Response

Authors’ Response to Reviewer’s Comments

The authors appreciate the suggestions and comments made by the reviewer. Modifications were made and highlighted in the revised version, according to the reviewer’s comments. In the following, the alterations made in the revision are summarized:

1. Firstly, on line 170 of page 7 the authors state that "the positive temperature coefficient of coercivity is supposed to be related to the magnetocrystalline anisotropy of MnBi"I found this statement quite vague as "supposed to be related" does not say anything. Can the authors explain how it is related? Is it for instance because the anisotropy constant itself increases as function of temperature or are there other factors at play? Please clarify how this can be explained.

Reply: owing to the non-uniform change of lattice constants with temperature [7] is added.

2. Secondly, although the manuscript is well written, some sentences have strange grammatical structures. Can the authors let the manuscript get proofreader by a native speaker (or does this journal do this upon acceptance) to edit the text to proper English?

Reply: English language (grammar and spelling) has been improved.

Reviewer 3 Report

The Authors present nice method to produce possible permanent magnetic material by different ball milling processes. They present significant influence of heating and surfactant on grain size and final value of coercivity.

I suggest to perform revision of the text: English language and once more read the text carefully. There are some mistakes in references to the figures, e.g. line 102.

Author Response

The authors appreciate the suggestions and comments made by the reviewer. Modifications were made and highlighted in the revised version, according to the reviewer’s comments. In the following, the alterations made in the revision are summarized:

1. I suggest to perform revision of the text: English language and once more read the text carefully. There are some mistakes in references to the figures, e.g. line 102.

Reply: English language (grammar and spelling) has been improved.

Figure 3 and 4 was changed to Figure 4 and 5.

Reviewer 4 Report

The manuscript has a good research potential and is worth publication after some minor corrections suggested below:

Abbreviations LTP, LEBM, SALEBM are explained in the Abstract only; usually the abbreviations are also explained in the body text, depending on the journal. However, unless there is a page limit, it is suggested to do so in the main text as well.

Page 2, line 59: “The sieved powders were milled in 2 methods” should probably be “The sieved powders were milled using 2 methods” or similar (English language)

Page 2, line 71: “…a speed of 40 m/s and subsequently annealing.” Should probably be “…a speed of 40 m/s and subsequently annealed” or similar (English language).

Page 2, lines 70-76: It is not clearly distinguished or described sufficiently whether ribbons, crushed ribbons and powders and in what state (as-cast, annealed) are used for XRD, e.g. line 71 mentions powders while line 74 mentions ribbons. Please distinguish clearly.

Page 3, lines91-92: please reformulate the statement to make it more clear.

Page 3, line 94: “…the Bi phase at 27.3..) should probably be “the reflection of Bi phase at 27.3…).

Page 5, lines 119-120 and Figs. 4 and 5, namely Fig.4d and Fig 5c; also Conclusions, page 8, line 176-177: The statement “…Compared Figure 4 with with Figure 5, at the same milling time, the powders size of 119 SALEBM is smaller than the size of LEBM.” is not evident from the provided images; rather, powder particles in Fig. 5c seem finer than those in Fig.4d.

While this argument is not crucial for the conclusions (except those on lines 176-177), it should be justified better, preferably quantified in detail or omitted/softened.

p. 5 line 129: ”... particle size may be smaller than the single domain size [23].”

Ref 23 (Rao et al, JALCOMM 2014) gives only sparse and contradictory information about this:

“…22 emu/g at 30 kOe and a high Hc of 16.3 kOe. Thus, even though the obtained magnetization in the nanoparticles is lower than that of the bulk alloys or micron sized particles [3,17], their Hc value is higher. Since the MnBi/Bi nanoparticles contain large volume fraction of non-magnetic Bi phase, a lower magnetization is observed.

The achieved large Hc is due to smaller particle size which is less than the single domain size of MnBi (500 nm).”

It is suggested to use a more proper reference to support the statement or omit the conclusion.

Add arrows (or us legend) to assign individual curves to left and right-hand axes in Figs. 6b, 7b, 8b to improve legibility.

Author Response

Authors’ Response to Reviewer’s Comments

The authors appreciate the suggestions and comments made by the reviewer. Modifications were made and highlighted in the revised version, according to the reviewer’s comments. In the following, the alterations made in the revision are summarized:

1. Abbreviations LTP, LEBM, SALEBM are explained in the Abstract only; usually the abbreviations are also explained in the body text, depending on the journal. However, unless there is a page limit, it is suggested to do so in the main text as well.

Reply: The abbreviations of LTP, LEBM, SALEBM and Ms were explained in the body text.

2. Page 2, line 59: “The sieved powders were milled in 2 methods” should probably be “The sieved powders were milled using 2 methods” or similar (English language)

Reply: “in 2 methods” was changed to using 2 methods”.

3. Page 2, line 71: “…a speed of 40 m/s and subsequently annealing.” Should probably be “…a speed of 40 m/s and subsequently annealed” or similar (English language).

Reply: “…a speed of 40 m/s and subsequently annealing.” was changed to …a speed of 40 m/s and subsequently annealed in vacuum”.

4. Page 2, lines 70-76: It is not clearly distinguished or described sufficiently whether ribbons, crushed ribbons and powders and in what state (as-cast, annealed) are used for XRD, e.g. line 71 mentions powders while line 74 mentions ribbons. Please distinguish clearly.

Reply: “…powders” was changed to ribbon powders”.

5. Page 3, lines91-92: please reformulate the statement to make it more clear.

Reply:  “…the main phase of Mn55Bi45 powders has become from the LTP MnBi phase ball to the Bi phase.” was changed to The Bi phase has become the main phase of Mn55Bi45 powders in Figure 3e.”.

6. Page 3, line 94: “…the Bi phase at 27.3..) should probably be “the reflection of Bi phase at 27.3…).

Reply: The reflection of Bi phase at 27.3….“ was added.

7. Page 5, lines 119-120 and Figs. 4 and 5, namely Fig.4d and Fig 5c; also Conclusions, page 8, line 176-177: The statement “…Compared Figure 4 with with Figure 5, at the same milling time, the powders size of SALEBM is smaller than the size of LEBM.” is not evident from the provided images; rather, powder particles in Fig. 5c seem finer than those in Fig.4d.

While this argument is not crucial for the conclusions (except those on lines 176-177), it should be justified better, preferably quantified in detail or omitted/softened.

Reply: We agreed that “…Compared Figure 4 with with Figure 5, at the same milling time, the powders size of SALEBM is smaller than the size of LEBM.” was not evident from the provided images. “…at the same milling time( < 14 h)…’ was added.

8. p. 5 line 129: ”... particle size may be smaller than the single domain size [23].”

Ref 23 (Rao et al, JALCOMM 2014) gives only sparse and contradictory information about this:

“…22 emu/g at 30 kOe and a high Hc of 16.3 kOe. Thus, even though the obtained magnetization in the nanoparticles is lower than that of the bulk alloys or micron sized particles [3,17], their Hc value is higher. Since the MnBi/Bi nanoparticles contain large volume fraction of non-magnetic Bi phase, a lower magnetization is observed.

The achieved large Hc is due to smaller particle size which is less than the single domain size of MnBi (500 nm).”

It is suggested to use a more proper reference to support the statement or omit the conclusion.

Reply: ”...a maximum, then decreases due to the particle size may be smaller than the single domain size [23].” was changed to “a maximum due to the smaller particle size is less than the single domain size”.

9. Add arrows (or us legend) to assign individual curves to left and right-hand axes in Figs. 6b, 7b, 8b to improve legibility.

Reply: Legend was added in Figures 6b, 7b and 8b
